# Cross-Talk between Amyloid, Tau Protein and Free Radicals in Post-Ischemic Brain Neurodegeneration in the Form of Alzheimer’s Disease Proteinopathy

**DOI:** 10.3390/antiox11010146

**Published:** 2022-01-11

**Authors:** Ryszard Pluta, Jacek Kiś, Sławomir Januszewski, Mirosław Jabłoński, Stanisław J. Czuczwar

**Affiliations:** 1Laboratory of Ischemic and Neurodegenerative Brain Research, Mossakowski Medical Research Institute, Polish Academy of Sciences, Pawińskiego 5 Str., 02-106 Warsaw, Poland; sjanuszewski@imdik.pan.pl; 2Department of Urology, 1st Military Clinical Hospital with the Outpatient Clinic, Al. Racławickie 23, 20-049 Lublin, Poland; jacekkis@gmail.com; 3Department of Rehabilitation and Orthopedics, Medical University of Lublin, Jaczewskiego 8 Str., 20-090 Lublin, Poland; mbjablonski@gmail.com; 4Department of Pathophysiology, Medical University of Lublin, Jaczewskiego 8b Str., 20-090 Lublin, Poland; stanislaw.czuczwar@umlub.pl

**Keywords:** brain ischemia, free radical, oxidative stress, reactive oxygen species, reactive nitrogen species, amyloid, tau protein, elastin-derived peptides, hormesis, vitagene, proteinopathy

## Abstract

Recent years have seen remarkable progress in research into free radicals oxidative stress, particularly in the context of post-ischemic recirculation brain injury. Oxidative stress in post-ischemic tissues violates the integrity of the genome, causing DNA damage, death of neuronal, glial and vascular cells, and impaired neurological outcome after brain ischemia. Indeed, it is now known that DNA damage and repair play a key role in post-stroke white and gray matter remodeling, and restoring the integrity of the blood-brain barrier. This review will present one of the newly characterized mechanisms that emerged with genomic and proteomic development that led to brain ischemia to a new level of post-ischemic neuropathological mechanisms, such as the presence of amyloid plaques and the development of neurofibrillary tangles, which further exacerbate oxidative stress. Finally, we hypothesize that modified amyloid and the tau protein, along with the oxidative stress generated, are new key elements in the vicious circle important in the development of post-ischemic neurodegeneration in a type of Alzheimer’s disease proteinopathy.

## 1. Introduction

The human brain accounts for only 2% of the total body weight, uses about 20% of the oxygen supplied to the whole body [1,2] and produces more free radicals than other organs of the body [2,3]. The brain’s enormous energy consumption makes it more susceptible to oxidative stress than any other organ in the human body [1,3]. The main structural and functional part of the brain, namely neurons, are particularly susceptible to oxidative injury due to the high level of metabolism compared to other cells of the nervous system [4]. The human brain is the most complex organ that controls all of the body’s responses. It contains billions of neurons that continuously control the proper functioning of the body. Neurons oversee and transmit signals that enable us to speak, think, move and achieve everything we do [5]. The brain’s neurons are closely interconnected to form a neural network. Therefore, the slightest disconnection between neurons can lead to disruption of various activities controlled by the brain, causing serious damage to the brain [5]. Disrupting the transmission of information between neurons in the brain has devastating effects, affecting human brain functions such as memory, movement, intelligence, speech and many other activities [5,6].

Human ischemic stroke is an age-related neurodegenerative disorder that is the leading cause of long-term disability and mortality, placing a huge burden on patients and their carers, the community and the healthcare system [7]. Currently available therapies for ischemic stroke are intravenous injection of tissue plasminogen activator and thrombectomy. However, both treatments are limited by a narrow therapeutic window and should be performed within 4.5 and 12 h, respectively, of the onset of an ischemic stroke. These limitations mean that the vast majority of post-ischemic patients are inadequately treated and therefore develop severe neurological deficits, with frequent full-blown dementia [8,9,10]. So, restoring blood flow following cerebral ischemia is an important challenge for stroke researchers and clinicians. Therefore, the effective restoration of blood circulation to the area of the ischemia is the main goal after the onset of the acute phase of an ischemic stroke. It should be emphasized that there may also be complications during the restoration of blood flow, among which one of the most serious further and complicated risks is the brain reperfusion itself. Restoration of blood flow is an inevitable problem once circulation is restored by therapy at the site of a stroke and carries with it important additional neuropathological changes. Despite the successful restoration of blood flow, neurological progress is still not observed in a significant number of patients due to the presence of neuropathological phenomena that prevent the return to normal activity [11,12]. After reversible cerebral ischemia with recirculation changes, now referred to as proteomic, genomic and biochemical cascades, they damage the ischemic parenchyma of the brain, counteracting the beneficial effects of restoring cerebral circulation [13,14]. During post-ischemic reperfusion, neuropathological pathways include excitotoxicity, intracellular Ca^2+^ accumulation, neuroinflammation, free radical and amyloid generation, modification of tau protein, necrosis, apoptosis, lipolysis, and elastin degradation to polypeptides [15,16,17,18,19]. Recently, elastin-derived polypeptides have been reported to induce overproduction of β-amyloid peptides in a model of Alzheimer’s disease [20,21,22]. Among these various neuropathological phenomena, free radical changes in brain tissue play an important role in the process of ischemic brain damage during recovery of blood flow at the primary ischemic focus.

The oxidation of nucleic acids, proteins and lipids in neurons is crucial for the survival or death of neuronal cells during recirculation at the area of cerebral ischemia. Neurons contain large amounts of polyunsaturated fatty acids that interact with reactive oxygen species, triggering a self-propelling cascade and initiating, for example, increased lipid peroxidation [23]. It should be emphasized that neuronal cells are low in glutathione, an essential antioxidant for scavenging free radicals [24]. Consequently, neuronal cells are extremely susceptible to oxidative stress. Accordingly, post-ischemic stroke patients are more predisposed to cognitive impairment and Alzheimer’s disease type dementia, which significantly affects the patient’s quality of life. All of these symptoms are related to secondary brain damage after stroke [25].

Oxidative stress indicators are not limited to ischemia of the injured brain [26], but are also present in serum or cells of the immune system and this is a universal phenomenon [27]. Therefore, in systemic blood circulation, after ischemia, elevated components of oxidative stress and mediators of chronic inflammation were found in patients, and in addition to the impaired immune response, polynuclear and mononuclear leukocytes were activated [26,28]. Leukocytes release, in addition to the overproduction of oxidation products, including reactive oxygen species and pro-inflammatory cytokines, including interleukin-6 or interleukin-1β [28]. As a result, the overproduction of reactive oxygen species through increased activity of a number of oxidative enzymes (e.g., myeloperoxidase, xanthine oxidase, etc.) or through dysregulation of the mitochondrial respiratory chain along with the elimination of antioxidants (e.g., catalase, superoxide dismutase, glutathione, glutathione peroxidase, etc.) finally causes oxidation of molecules (e.g., DNA, lipids and proteins), which ultimately damage cells and tissues. Moreover, since oxidation and inflammation are linked phenomena [26], the accumulation of oxidative factors trigger the release of pro-inflammatory mediators, and then strengthen and continue the vicious cycle of oxidative damage, inducing progressive neurodegeneration of the brain tissue [28].

Recently, an intensively researched post-ischemic neuropathology has emerged in connection with genomic and proteomic development, which has transferred ischemia-reperfusion brain injury to a new level of neuropathological mechanisms, due to the presence of amyloid plaques and the development of neurofibrillary tangles, which further exacerbate oxidative stress in the brain in addition to the classic processes described so far after ischemia. The major amyloid plaque molecule are β-amyloid peptide 1-42 or 1-40, which is derived from proteolytic metabolism of the amyloid protein precursor. Additionally hyperphosphorylated tau protein is the major component of neurofibrillary tangles. Therefore, oxidative stress is a common and key factor in post-ischemic brain injury during recirculation. It consists of an increased generation of reactive oxygen species and reactive nitrogen species that damage various components of the cell such as DNA, lipids and proteins [29]. DNA oxidative injury is one of the deadliest end effects of increased oxidative stress post-ischemia [30,31]. Oxidative damage to DNA occurs within minutes in an ischemic brain [30,31], but oxidative breakdown can continue for up to six months after a stroke [27]. Problems with repairing damaged DNA result in the activation of various mechanisms that trigger the death of neuronal cells through apoptosis that threatens functional regeneration after stroke, with a high probability of developing full-blown dementia [31].

Finally, in this review we will discuss the hypothesis that modified amyloid with tau protein hyperphosphorylation and oxidative stress are new key elements in the vicious cycle important in post-ischemic neurodegeneration with Alzheimer’s disease-type proteinopathy. In addition, we will present a new look at the oxidative stress generated by amyloid and tau protein, and at the same time show the effect of oxidative stress on the behavior of amyloid and tau protein in the context of post-ischemic brain neurodegeneration with the development of Alzheimer’s disease-type proteinopathy. The data show a progressive large redox imbalance due to the overproduction and accumulation of oxidative factors in the cells and brain tissue, which exceeds the clearance capacity by the antioxidant system. We will also focus on the role of two main factors, such as amyloid and tau protein, which trigger oxidative stress in post-ischemic reperfusion injury in the vicious cycle.

## 2. Free Radicals in Recirculation

Free radicals fall into two main groups: reactive oxygen species and reactive nitrogen species [2]. Reactive oxygen species and reactive nitrogen species play a key role in many pathological processes, including ischemic-reperfusion injury. Therefore, the toxicity of free radicals in ischemic-reperfusion injury is currently being intensively studied. Based on the above data, the concept of “oxidative stress” was introduced by Helmut Sies [32] as an imbalance between the production of oxidants and antioxidants that can cause damage to various organs and organisms. Since then, the field of redox biology has developed the concept of the important role of oxidative stress in the development of various pathologies, including cerebral ischemia [33,34]. In the following sections, we will discuss some of the mechanisms involved in the toxicity of free radicals to the brain during ischemia-reperfusion injury.

### 2.1. Reactive Oxygen Species in Recirculation

Oxidative stress is triggered by the overproduction of reactive oxygen species. The main types of reactive oxygen species include superoxide anion, hydroxyl radicals and hydrogen peroxide. Under normal circumstances, superoxide dismutase, glutathione peroxidase, catalase, and other antioxidants protect the brain from oxidative stress. In the following, we will discuss some of the mechanisms involved in the toxicity of reactive oxygen species to brain tissue during post-ischemic reperfusion [2]. During post-ischemic reperfusion, the production of free radicals increases significantly, leading to the breakdown of the antioxidant processes [2]. According to previous research, overproduction of reactive oxygen species involves four pathways: NADPH oxidase, mitochondrial respiratory chain dysfunction, xanthine reaction mediated by xanthine oxidase and arachidonic acid catalyzed by cyclooxygenase 2 [2,35]. In the early stages of cerebral ischemia, reactive oxygen species are mainly produced by the mitochondria [2].

Following ischemia, the brain lacks oxygen and glucose, which negatively affects ATP production. In the case of ATP deficiency, calcium levels in neurons increase, causing massive production of reactive oxygen species through the depolarization of the mitochondria [2,36,37,38]. Along with the penetration of macrophages and other immune cells into the brain during the development of neuroinflammation, the activation of NADPH oxidase in these cells contributes to the production of reactive oxygen species [39]. With recirculation of the blood, an abundance of oxygen appears which causes oxidative damage. It is well known that oxidative stress can activate pro-apoptotic mechanisms such as the cytochrome c pathway, autophagy, inducing DNA damage, affecting protein structure and function, especially lipid peroxidation during reperfusion [40,41,42].

### 2.2. Brain Damage Due to Reactive Oxygen Species during Recirculation

Oxidative stress can induce cell death by modifying the structure and function of proteins, DNA damage and lipid peroxidation. DNA injury includes active and passive DNA damage. Oxidative stress causes particularly passive DNA injury. Active DNA injury (mediated by DNA endonucleases, which mainly contain caspase-activated deoxynuclease, endonuclease G, and an apoptosis-inducing factor) causes double-stranded DNA fragmentation (Figure 1). Passive DNA injury is caused by the direct reaction of DNA with reactive oxygen species or the indirect reaction of DNA with the products of the reaction of reactive oxygen species with proteins or lipids, leading to modification of nucleotide bases or the development of single or double strand breaks (Figure 1) [2,31].

The hydroxyl radical, a form of reactive oxygen species produced by the Fenton reaction, can cause lipid peroxidation. As a result of the reaction of hydroxyl radicals with unsaturated fatty acids, an alkyl radical is formed, which when reacted with molecular oxygen, can form a peroxyl radical. Then the peroxyl radical receives hydrogen from another fatty acid and produces lipid hydroperoxide and a second alkyl radical, which causes another lipid peroxidation [2]. Lipid peroxidation damages membranes, leading to organelle dysfunction and changes in ion transport [2,43]. Lipid peroxidation products play an important role in damage from oxidative stress. These products are 4-hydroxynonenal, malondialdehyde and acrolein [2]. They can cause protein dysfunction, leading to more dangerous damage from oxidative stress [2]. Protein 53 is a key molecule in the process of generating reactive oxygen species during the death of neurons in the brain following ischemia [2,44,45]. Reactive oxygen species can cause a protein 53 reaction with cyclophilin D, which opens the mitochondrial transition pores, causing mitochondrial swelling and cells necrosis (Figure 2) [2,46]. Reactive oxygen species increase the permeability of the mitochondrial membrane and affect the released cytochrome c, which forms an inhibitory complex with proteins of the Bcl-2 family, such as Bid and Bax, and protein 53. Cytochrome c can generate active caspases by forming a complex with procaspase-9, an apoptotic protease activating factor-1 and ATP, resulting in apoptosis of pyramidal neurons in the CA1 region of the hippocampus following reversible global cerebral ischemia in rats (Figure 2) [2,47]. Protein 53 apoptosis modulator is a key pro-apoptotic protein that belongs to the Bcl-2 protein family [2,48]. Some evidence suggests that inhibition of oxidative stress by abundant superoxide dismutase 1 may suppress upregulated apoptosis modulator by protein 53, indicating an association between protein 53 upregulated apoptosis modulator and oxidative stress in death of pyramidal neurons in the CA1 region of the hippocampus after reversible global ischemic brain injury [2,45]. Another most important cell death regulating mechanism is related to the mitogen-activated protein kinase, which is associated with the activity of reactive oxygen species. Mitogen-activated protein kinase has been shown to induce the death of neuronal cells in the cortex and hippocampus in transient cerebral ischemia in mice [2,49]. The mechanism of the mitogen-activated protein kinase consists of three main components: extracellular signal-regulated kinase 1/2, c-Jun NH2-terminal kinase, and protein 38 mitogen-activated protein kinase. C-Jun NH2-terminal kinase and protein 38 mitogen-activated protein kinase play an important role in apoptosis. The mechanisms of the c-Jun NH2 terminal kinase and the protein 38 mitogen-activated protein kinase can be activated by the apoptosis signal regulating kinase 1, which is activated by reactive oxygen species, inducing post-ischemic apoptosis [2]. Additionally, as post-ischemic neuroinflammation develops, immune cells containing NADPH oxidase produce significant amounts of reactive oxygen species, worsening damage from oxidative stress [2]. Also, reactive oxygen species can activate inflammatory cells. Reactive oxygen species activate neutrophils, microglia and macrophages by the nuclear factor kappa B pathway. Leukocytes possess myeloperoxidase that can trigger the production of hydrochloric acid, a strong oxidant that is essential for the survival and plasticity of neuronal cells post-ischemia [2,50].

### 2.3. Reactive Nitrogen Species in Recirculation

There are two specific types of reactive nitrogen species, oxidative peroxynitrite and NO, which dominate the progression of post-ischemic brain injury [2]. NO is produced by the reaction of oxygen with L-arginine, which is controlled by three types of nitric oxide synthase, including neuronal nitric oxide synthase, endothelial nitric oxide synthase and induced nitric oxide synthase. Neuronal nitric oxide synthase and endothelial nitric oxide synthase are calcium dependent, while induced nitric oxide synthase is calcium independent. Overall, the NO generated from induced nitric oxide synthase and neuronal nitric oxide synthase is harmful to cells. It was shown that mice lacking endothelial nitric oxide synthase had a greater infarct volume than control mice in post-ischemic brain injury [2,51]. Endothelial nitric oxide synthase produces huge amounts of NO at the onset of ischemia in endothelial cells, causing the blood vessels to widen [2]. In contrast, NO, produced mainly from activated induced nitric oxide synthase and neuronal nitric oxide synthase, is detrimental to focal ischemic brain injury [2]. In addition, it has been shown that overactive endothelial nitric oxide synthase may be harmful, and its limited inhibition may act preventively in post-ischemic conditions by inhibiting the development of peroxynitrite [2,52]. Additionally, NO reacts with O^2−^ and generates a strong oxidant and peroxynitrite, which is associated with a stronger oxidation than both O^2−^ and NO together [2,53].

### 2.4. Brain Damage Due to Reactive Nitrogen Species during Recirculation

Too much NO can lead to loss of control of the blood-brain barrier, neuroinflammation, and death of neurons [2]. Both matrix metalloproteinase and tight junction proteins are influenced by NO, which opens the blood-brain barrier after ischemia [54,55,56]. NO stimulates matrix metalloproteinase-2 in the first phase, and in the second phase, it activates matrix metalloproteinase-9, which affects the opening of the blood-brain barrier [2]. Activated matrix metalloproteinases disintegrate the extracellular matrix and tight junction proteins of the walls of blood vessels [57]. Excess NO causes neuronal death by protein modification, mitochondrial dysfunction and the formation of peroxynitrite [2]. During brain ischemia, NO strongly inhibits cytochrome c oxidase in the respiratory chain of the mitochondria [2]. In contrast, the reaction of NO with proteins leads to the nitrosylation of proteins or the formation of nitrosothiol [2,58]. NO also increases the activity of cyclooxygenase-2, which can mediate glutamate toxicity by generating additional oxygen reactive species and participate in the neuroinflammatory response by pro-inflammatory production of prostaglandin E2 [59]. Serious damage to the brain after ischemia by oxidative peroxynitrite is the death of neurons and changes in the permeability of the blood-brain barrier. The reaction of the oxidant peroxynitrite with tyrosine produces 3-nitrotyrosine, which causes damage to the proteins of the cytoskeleton and inhibition of enzymatic activity, resulting in the death of neurons [2,60]. In addition, the oxidant peroxynitrite reacts with DNA elements, such as the sugar phosphate backbone and guanine nucleotides, causing DNA damage and ultimately activating poly (adenosine diphosphate-ribose) polymerase (PARP) processes [2]. Increased PARP-1 activation depletes NAD^+^, leading to cell death, including neuronal cells [2,61]. In addition, oxidative peroxynitrite causes mitochondrial dysfunction by affecting the I-V complexes of the mitochondrial respiratory chain [2,62,63]. Peroxidation of membrane lipids by oxidative peroxynitrite also causes cells death [2,64]. There are also observations that oxidative peroxynitrite can activate matrix metalloproteinase-1, -2 and -9, leading to dysfunction of the tight junction proteins of blood vessels [65,66], finally triggering the permeability of the blood-brain barrier after ischemia-reperfusion injury to the brain [2].

## 3. Amyloid Accumulation in Post-Ischemic Brain

The key processes related to amyloid accumulation and its effects on ischemic brain injury are virtually unclear. The following is the amyloidogenic processing of the post-ischemic amyloid protein precursor that is associated with the production and deposition of amyloid in the brain following ischemia [67,68,69]. In addition, it has been shown that patients with a history of ischemic stroke have an increase in elastin-derived polypeptides in the cerebrospinal fluid that contribute to the accumulation of amyloid in the extracellular space [19,20,21,22]. The accumulation of amyloid in the brain after ischemia [67,70] and its presence in the blood [71,72] indicate an important role of amyloid in the progression of ischemic brain injury with Alzheimer’s disease proteinopathy.

The expression of the *amyloid protein precursor* (*APP*) gene in the CA1 area of the hippocampus decreased within 2 days, in contrast, it increased 7–30 days after ischemia (Table 1) [68]. Expression of the *β-secretase* gene (*BACE1*) increased during 2–7 days and decreased 30 days following ischemia (Table 1) [68]. *Presenilin 1* and *2* (*PSEN1* and *2*) genes increased in 2–7 days and decreased in 30 days after ischemia (Table 1) [68].

In the CA3 region of the hippocampus, the expression of the *APP* gene fluctuated around the control values within 2 and 30 days after ischemia (Table 1) [69]. But 7 days post-ischemia, expression of the above gene increased [69]. The expression of the *α-secretase* (*ADAM10*) gene fell below control during 2, 7, 30 days post-ischemia (Table 1) [69]. Expression of the *BACE1* gene in the above region decreased in 2–7 days and increased 30 days after ischemia (Table 1) [69]. The *PSEN1* gene was elevated within 2–7 days and was fluctuating around control 30 days following ischemia (Table 1). In this area, post-ischemic *PSEN2* gene expression fluctuated around control within 2 days, while it decreased on day 7, but increased above control on day 30 (Table 1) [69].

The expression of the *APP* gene in the temporal cortex decreased within 2 days of survival and increased above the control 7–30 days after ischemia (Table 2) [73]. The highest increase in *BACE1* gene expression was recorded within 2 days, but 7–30 days after ischemia, it oscillated around the control (Table 2) [73]. *PSEN1* gene expression fluctuated about control within 2, 7 and 30 days post-ischemia (Table 2) [74]. The highest increase in the expression of the *PSEN2* gene was observed within 2 days, while in 7–30 days after ischemia, the expression of this gene oscillated around the control (Table 2) [74].

After transient experimental cerebral ischemia with a survival of up to 2 years, amyloid accumulation in the intra- and extracellular space was observed [67,75,76,77,78,79,80,81,82,83,84,85,86]. Accumulation of amyloid in the extracellular space has been observed as diffuse or senile amyloid plaques [67,76,82,85,87,88,89,90,91,92,93]. Amyloid plaques have been found in the hippocampus, cerebral cortex, thalamus, around the lateral ventricles, and in the corpus callosum. The accumulation of amyloid inside neuronal and neuroglial cells indicates the importance of amyloid in the progressive neurodegeneration of the brain after ischemia with the development of Alzheimer’s disease proteinopathy [67,80,87,88,94,95,96,97,98]. These data indicate that after experimental ischemia, amyloid production is responsible for additional neurodegenerative processes that may worsen post-ischemic neurological outcomes due to the constant wasting of neurons [7,76,82,85,86,99,100]. Amyloid is a toxic molecule that activates intracellular mechanisms in ischemic astrocytes, neurons and microglia, which further induce damage and loss of neuronal and glial cells following ischemia. Following ischemia in the human brain, both diffuse and senile amyloid plaques have been observed in the brain cortex and hippocampus [70,101,102,103]. In addition, clinical trials showed that blood amyloid was increased in patients with post-ischemic brain damage [71,72]. The rise in blood amyloid negatively correlated with the neurological outcome after cerebral ischemia [72].

## 4. Amyloid versus Oxidative Stress in Recirculation

Studies have shown that methionine 35 from β-amyloid peptide is important for oxidative stress and its toxicity induced by amyloid [104,105,106]. Methionine undergoes two-electron oxidation to form methionine sulfoxide [106,107]. Oxidation of methionine to sulfoxide plays a significant role in reducing the antioxidant defense of cells [108]. In addition, the presence of methionine sulfoxide reductase suggests that sulfoxide reductase may influence the antioxidant effect [106,109,110,111]. Additionally, methionine residing in side amyloid in the lipid bilayer can undergo single-electron oxidation to form a sulfuranyl free radical. Since amyloid is generated from an amyloid protein precursor, it is suggested that the amyloid once produced may insert as small oligomer into the lipid bilayer adopting the α-helical conformation [106]. This suggests that the secondary structure of amyloid is a peptide neurotoxicity factor. It has been suggested that molecular oxygen or Cu^2+^ may be the key to the oxidation of methionine to the sulfuranyl radical. In the absence of oxygen, amyloid cannot lead to the formation of free radicals [106]. In vitro studies have shown that β-amyloid peptide can promote the reduction of Cu^2+^ to Cu^+^ by binding to Cu^2+^ and forming hydrogen peroxide. Then, the Cu^+^ can react with hydrogen peroxide to form highly reactive hydroxyl free radicals [106,112]. The importance of copper in amyloid-induced toxicity is suggested by a study where methionine 35 was replaced with valine, which showed an increase in toxicity, suggesting that this substitution may change the amyloid conformation from α-helix to a mixture of α-helical and β-sheet conformations, thus increasing Cu^2+^ binding followed by associated toxicity [106]. In addition, the replacement of histidine 6, 13 and 14 in the β-amyloid peptide 1–42 by tyrosine, which binds to Cu^2+^ with a lower affinity than histidine, showed that this did not affect oxidative stress and neurotoxicity, further emphasizing the importance of methionine in inducing oxidative stress and its toxicity by amyloid [106].

After generating the sulfuranyl free radical, it can detach allylic H atoms from the acyl chains of unsaturated fatty acids in the lipid bilayer to initiate the lipid peroxidation process and, consequently, affect the lipid bilayer [106]. The oxidation products further diffuse across the membrane, affecting other cellular organelles, greatly enhancing the action of the primary amyloid free radical, leading to neuronal cells death and post-ischemic brain neurodegeneration.

Studies with transgenic Caenorhabditis elegans expressing human β-amyloid peptide 1-42 presented increased oxidation which correlated with worm paralysis [106,113]. However, some studies have found that the 33-35 area of the β-amyloid peptide 25-35 is critical to the aggregation and neurotoxic properties of the β-amyloid peptide [106]. Study in transgenic mice presented that methionine residing in the β-amyloid peptide affects neurotoxicity, oxidative stress, and the formation of amyloid plaques [114].

## 5. Tau Protein Modification in Post-Ischemic Recirculation

Recent studies have documented an association between the expression of the *tau protein* (*MAPT*) gene in the CA1 and CA3 regions of the hippocampus after ischemia with a 30-day survival (Table 1) [69]. In the CA1 area, the expression of the *MAPT* gene increased significantly during 2 days following the ischemic brain injury (Table 1). In contrast, expression of the above gene was reduced 7–30 days post-ischemia (Table 1). After ischemia in the CA3 area, an oscillation of the expression of the *MAPT* gene around the control was observed 2 days after ischemia (Table 1) [69]. In contrast, the expression of the above gene increased 7–30 days after ischemia (Table 1) [69].

Studies of ischemic brains in animals and humans have documented the accumulation of tau protein in neurons and glial cells in the hippocampus, cortex, and thalamus [83,115,116,117,118,119,120,121]. An increase in soluble tau protein in brain tissue using microdialysis in an ischemic brain is also shown [122]. In addition, studies have indicated that the modified tau protein inhibits the movement of neurofilaments, organelles, vesicles with an amyloid protein precursor, and increases oxidative stress in neurons [123].

An increase in blood tau protein level was observed after global cerebral ischemia in humans with a maximum increase on days 2 and 4 of survival [124]. This observation correlated with two types of neuronal death in the brain: necrosis and apoptosis, respectively [125]. The above information suggests that blood tau protein levels may be a prognostic factor for neurological outcome post-ischemia [124,125]. Increased levels of tau protein have also been found in the blood of patients after ischemic stroke, which is probably a factor in the progression of damage in neuronal cells [126]. In addition, an increase in tau protein has been observed in humans in the cerebrospinal fluid following an ischemic stroke [126]. Moreover, post-ischemia, blood-derived tau protein [126] crosses the ischemic blood-brain barrier in two directions and can enhance its own pathology in the brain [127]. Finally, ischemic blood-brain barrier insufficiency may exacerbate tau protein neuropathology in the brain and also suggests that post-ischemic brain injury may be part of the cause responsible for the increase in plasma tau protein concentration [126,127].

After transient local ischemic damage to the brain, the tau protein underwent excessive hyperphosphorylation and accumulated at the border of the ischemic focus [128]. Following forebrain ischemia in gerbils, tau protein phosphorylation has been shown to be controlled by mitogen-activated protein kinase, cyclin-dependent kinase 5, and glycogen synthase kinase 3 [129]. New data show that after ischemia, the tau protein undergoes hyperphosphorylation in neuronal cells and accompanies their apoptosis [120,121,128,130,131].

The information presented argues that the tau protein undergoes continuous hyperphosphorylation after a transient episode of cerebral ischemia. This triggers post-ischemia the generation of paired helical filaments [132], neurofibrillary tangle-like [128,130,131] and neurofibrillary tangles [133,134]. This points to the involvement of various modifications of the tau protein in the death of neurons in the brain following ischemia. The above evidence also explains a different principle regarding the time of death of neuronal pyramidal cells in the CA1 and CA3 areas of the hippocampus after ischemia in a manner dependent on tau protein hyperphosphorylation and modification of its structure.

## 6. Tau Protein versus Oxidative Stress in Recirculation

Several studies have documented that oxidative stress triggers tau protein phosphorylation in neuronal cells in vitro [135,136,137]. Additionally, carbonyl-4HNE facilitates hyperphosphorylation of the tau protein and aggregation in cultures [137,138,139,140]. In addition, oxidative stress also generates the oxidation of fatty acids which have been found to promote tau protein polymerization in cultures [137,141]. In mice deficient in mitochondrial superoxide dismutase 2, tau protein hyperphosphorylation, oxidative stress, and mitochondrial dysfunction were observed in parallel [137,142]. In addition, mitochondrial oxidative stress has been shown to induce hyperphosphorylation of the tau protein [137,142]. It was shown that zebrafish deficient in folic acid induced oxidative stress related to hyperphosphorylation of tau protein and its aggregation and accumulation of amyloid plaques [137,143,144]. Damage to mitochondria, development of oxidative stress, increased tau protein phosphorylation and nuclear translocation of glycogen synthase kinase 3 beta were observed in rabbits after intracisternal injection of β-amyloid peptide 1-42. It was shown that in vitro treatment of rat cortical neurons with cuprizone, a copper chelator, with Fe^2+^ and H_2_O_2_ oxidizing molecules, significantly increased the activity of glycogen synthase kinase 3 beta and tau protein hyperphosphorylation [137,145]. However, simultaneous addition of lithium chloride to these cultures lowers the activity of glycogen synthase kinase 3 beta and tau protein phosphorylation, recognizing glycogen synthase kinase 3 beta as an enzyme involved in tau protein phosphorylation after oxidative stress in this model [137,145]. This was supported by the promotion of the hyperphosphorylation of the tau protein observed in rat cortical neurons in vitro by glycogen synthase kinase 3 beta and oxidative stress [137,145]. Taken together, this evidence suggests that glycogen synthase kinase 3 beta plays an important role in the pathology of tau protein and that effective modulation of its action may prevent apoptosis and tau protein phosphorylation induced by oxidative stress [137,146]. However, in addition to the glycogen synthase kinase 3 beta, oxidative stress affects other signaling pathways and kinases that mediate tau protein hyperphosphorylation. In fact, 4-hydroxynonenal has also been found to directly activate stress-activated kinases, such as c-Jun NH2-terminal kinase and protein 38, in neurons in vitro [147]. Another possible element in the interaction between oxidative stress and tau protein phosphorylation is peptidyl prolyl cis-trans isomerase 1. Therefore, it has been found that the accumulation of hyperphosphorylated tau protein promotes the production of reactive oxygen species and oxidative stress (Figure 1 and Figure 2) [137]. Additionally, this evidence strongly suggests that oxidative stress, in turn, directly induces hyperphosphorylation of the tau protein [137]. In this context, tau protein hyperphosphorylation and oxidative stress emerge as two components of a vital “vicious circle” leading to a progressive increase in radical oxygen species and modified tau protein, and ultimately to neuronal death [137]. The data presented above clearly prove that the tau protein in patients and animals after cerebral ischemia causes oxidative stress in neuronal cells [137]. On the other hand, the line of evidence also suggests that the generation of reactive oxygen species may directly induce hyperphosphorylation and aggregation of the tau protein.

## 7. DNA Damage in Brain Cells during Recirculation

Ischemic brain damage leads to an increase in free radicals oxidative stress, which causes damage to DNA, gray and white matter and the blood-brain barrier. Oxidative damage to DNA occurs immediately after cerebral ischemia [30]. Oxidative damage to DNA in the brain following ischemia has been studied frequently in neurons. Unlike neurons, microglia, astrocyte, oligodendrocyte, and endothelial cells, which are also prone to ischemic damage, glial DNA damage is less studied than neuronal DNA damage.

### 7.1. Neurons

Research suggests that post-ischemic brain injury produces base modifications, apurinic/apyrimidinic sites, double- and single-strand breaks of DNA in neuronal cells, all of which activate pro-death signaling pathways and accelerate neuronal death (Figure 1) [148]. Study found that the amount of urine base modification was negatively correlated with neurological outcome after brain ischemia and significantly decreased after rehabilitation [148]. This information suggests that an increased amount of base modification in urine could be used as a biomarker for neurological outcome post-ischemia, while a decreased amount of base modification after rehabilitation is associated with a better prognosis. This evidence supports the idea that damage to DNA structure, for example the development of base modification, may serve as a prognostic marker after ischemia and that DNA repair may be a potential therapeutic target post-ischemia.

### 7.2. Microglia

Microglia cells are responding rapidly to cerebral ischemia, including production reactive oxygen species [2,148]. After ischemic damage, microglia shows oxidative DNA damage, including the generation of base modifications [31]. Microglia cells in mice can increase oxidative stress by the voltage-gated proton channel, thus contributing to DNA damage [148]. It should be noted that microglia plays a dual role in relation to cerebral ischemia due to different phenotypes, namely with a pro-inflammatory phenotype it threatens the life of neurons and with a repair phenotype it supports the survival of neurons [29,31,148]. Thus, modulation of the microglia phenotype may have a favorable effect on post-ischemia recovery.

### 7.3. Astrocytes

Astrocytes are also sensitive to DNA oxidative damage. Cerebral ischemia is known to induce astrocyte activity, and this phenomenon can cause DNA damage in astrocytes [148]. Activated astrocytes in the ischemic focus show oxidative damage to DNA, such as single- and double-strand breaks and base modifications (Figure 1) [148]. The activity of astrocytes influences the formation and transformation of the glial scar, the structure and function of neuronal cells and the blood-brain barrier after ischemia. Given the pivotal role of astrocytes in repairing blood-brain barrier, they are expected to facilitate the rebuilding of the blood-brain barrier and improve neurological outcomes following cerebral ischemia.

### 7.4. Oligodendrocytes

Since oligodendrocytes determine myelination, damage to the DNA in these cells can cause axonal demyelination following ischemia [148,149]. DNA damage and repair in oligodendrocyte cells has been demonstrated in studies after ischemic brain injury [149,150].

### 7.5. Endothelial Cells

DNA oxidative damage is also documented in the vascular endothelium. Experimental studies prove that endothelial cells in small vessels of the brain show signs of DNA single-strand breaks within 3 days after focal transient cerebral ischemia [148]. DNA damage in the vascular endothelium causes cognitive impairment, the development of cerebral edema and infiltration of inflammatory cells, which may directly affect the neurological outcome after cerebral ischemia [148,151].

## 8. Side Effects of Post-Ischemic Recirculation

Recirculation triggers serious complications, such as failure of the blood-brain barrier, development of cerebral edema and hemorrhagic transformation of the ischemic focus and the generation of free radicals. Patients who develop life-threatening complications have poorer neurological outcomes that can often be fatal.

### 8.1. Blood-Brain Barrier Insufficiency

Matrix metalloproteinases, a key factor in damage to the blood-brain barrier (Figure 3), can cause oxidative damage to DNA, resulting in neuronal death [148]. Brain ischemia may induce abnormal behavior of gelatinase A and B, which results in increased permeability of the blood-brain barrier (Figure 3) and enlargement of the ischemic focus [54,148]. Gelatinases, which cause damage to the blood-brain barrier, also act as nucleases that enter the nucleus of the neuron and break down nuclear DNA repair proteins in post-ischemic neuronal cells [148]. This evidence supports the existence of internal DNA damage that induces the death of neuronal cells following ischemia [148].

### 8.2. Brain Edema Development

Edema development induced by free radical oxygen stress includes cytotoxic and vasogenic edema (Figure 3). Cytotoxic edema is associated with abnormal ion transport across membranes. Ion transport proteins, which are oxidized by reactive oxygen species, are involved in ion pumps, ion channels, ion exchangers and ion cotransporters. Reactive oxygen species oxidize sulfhydryl groups located on ion transport proteins, peroxidate membrane phospholipids, reduce the amount of ATP and inhibit oxidative phosphorylation, which causes abnormal ion transport, causing cytotoxic edema (Figure 3) [152,153]. In addition, reactive oxygen species are involved in inhibiting glutamate uptake by Na^+^/glutamate transport [2]. The enormous release of glutamate during reperfusion injury [15] destroys K^+^, Na^+^ and Ca^2+^ homeostasis, causing membrane dysfunction and development of cytotoxic edema (Figure 3) [154]. Vasogenic edema is caused by a failure of the blood-brain barrier, and the likely mechanisms leading to damage to the blood-brain barrier are described above. (Figure 3).

### 8.3. Hemorrhagic Transformation of the Ischemic Focus

Hemorrhagic transformation is directly related to uncontrolled permeability of the blood-brain barrier for plasma and blood cells to the brain (Figure 3). The blood-brain barrier consists of endothelial cells, basement membrane, tight junctions, astrocytes and pericytes [2,155]. These cells have enzymes such as nitric oxide synthase and NADPH oxidase [2,52,156,157,158]. Following ischemia-reperfusion injury and treatment with thrombectomy or thrombolysis, H_2_O_2_ generated by NADPH oxidase induces changes in the blood-brain barrier by altering tight junction proteins and thereby increasing monocyte migration [2,159,160,161]. Free radicals, especially NO and peroxynitrite, activate the mechanisms of matrix metalloproteinase’s, which lead to the degradation of laminin and collagen in the basement membrane, which results in uncontrolled permeability of the blood-brain barrier after cerebral ischemia [157,161,162,163,164,165,166]. Matrix metalloproteinases have been shown to be involved in the development of hemorrhage associated with plasminogen activator [2,167]. Thus, it is possible that NO is involved in the hemorrhagic conversion of the ischemic brain following plasminogen activator therapy mediated by matrix metalloproteinases. In addition, plasminogen activator may increase matrix metalloproteinases, especially matrix metalloproteinase-9, via the lipoprotein receptor mechanism, but the ultimate effect of NO and plasminogen activator on hemorrhagic conversion following embolic ischemia of brain is still unknown [54,167,168,169,170]. It has been shown that peroxynitrite can inactivate plasminogen activator, influencing the course of action of thrombolysis [2,171,172,173].

## 9. Conclusions

Free radicals are physiological byproducts of cellular metabolism and are therefore inevitable. On the other hand, the imbalance between pro-oxidative and antioxidant cellular molecules leads to a vicious cycle where free radical oxidative stress causes oxidation of DNA, lipids and proteins in neurons and ultimately their death after ischemia. Oxidation can be considered a neuropathological marker in cerebral ischemia. It contributes to disease progression by increasing amyloid accumulation, tau protein hyperphosphorylation, and loss of neurons and synapses, leading to post-ischemic brain neurodegeneration in the form of Alzheimer’s disease proteinopathy (Figure 4). Additionally, several lines of evidence indicate that amyloid itself causes oxidative stress (Figure 4). Amyloid-triggered oxidative stress occurs in a bilayer in which the β-amyloid peptide is inserted as oligomer and serves as a source of reactive oxygen species [106].

It is worth noting that oxidative stress can modify the tau protein by phosphorylation which mediates the neurodegeneration e.g., in Drosophila melanogaster [174]. Phosphorylation of the tau protein blocks the movement of organelles, neurofilaments and vesicles of the amyloid protein precursor in neurons and increases oxidative stress post-ischemia (Figure 4) [123]. Oxidative stress is an important factor contributing to the accumulation and hyperphosphorylation of tau protein with the development of neurofibrillary tangles, indicating that it plays a key role in the pathogenesis of post-ischemic brain neurodegeneration in the form of Alzheimer’s disease proteinopathy (Figure 4). Oxidative stress has been shown to stimulate tau protein hyperphosphorylation and amyloid production, but the accumulation of amyloid and hyperphosphorylated tau protein also induces oxidative stress. Moreover, the close relationship between amyloid, hyperphosphorylated tau protein and oxidative stress suggests that this phenomenon is a key element in the vicious cycle that plays a significant role in the pathological process of modifying tau protein and amyloid, resulting in post-ischemic neurodegeneration of the brain in the form of Alzheimer’s disease proteinopathy (Figure 4).

In addition to the accumulation of free radicals, deficiency of superoxide dismutase, glutathione peroxidase and catalase has also been found after cerebral ischemia. Free radicals, especially reactive oxygen species and reactive nitrogen species, have intense oxidation or nitrification abilities in the brain. During post-ischemic brain recirculation with re-flow of blood, the massive production of reactive oxygen species and reactive nitrogen species leads to the death of neurons through DNA damage, protein dysfunction and lipid peroxidation. Oxidative/nitrosative stress in brain ischemia-reperfusion injury also plays a key role in inducing hemorrhagic transformation of the ischemic focus in conjunction with changes in the blood-brain barrier and development of cerebral edema (Figure 3). Research indicates that mitochondrial dysfunction [25,175], metal dyshomeostasis [41,176,177,178], tau protein hyperphosphorylation [123,174], neuroinflammation and amyloid accumulation [106] are the primary processes that induce oxidative stress. Inactivation and deficiency of antioxidant enzymes reduces the removal of free radicals from the brain during recirculation. This indicates that oxidative stress is an important neuropathological hallmark of the post-ischemic brain [175,179].

Broadly speaking, in order to survive cerebral ischemia, neuronal and glial cells developed integrated responses, so-called longevity safety processes, consisting of several genes called vitagenes, including but not limited to members of the heat shock protein system to detect and control various forms of pathological processes [180]. All of these considerations strongly support the idea that the smooth functioning of the maintenance and repair processes is crucial for the survival of brain cells under conditions of oxidative and nitrosative injury, and that tolerance to brain stress can be achieved through modulation of a multigene system such as the hormesis-vitagene network [180]. The efficient functioning of repair and maintenance processes is crucial for both survival and quality of life after ischemic brain injury. This is accomplished by a complex network of so-called longevity processes, which are made up of several genes of vitagene network, including the heat shock system, a highly conserved mechanism responsible for the repair and preservation of cellular macromolecules such as proteins, RNAs and DNA. During inflammation after cerebral ischemia, there is a gradual decline in the strength of the heat shock response, which may prevent protein damage from repairing, leading to the death of neuronal cells. In this situation, the concept of dose-response hormetic neuroprotection should be applied with the key role of heat shock proteins as elements of the vitagene network in neuroprotection and redox proteomics as redox modulators of stress-responsive vitagenes [181,182]. The use of hormetic machinery makes it possible to apply specific cell, tissue and/or pathological pathways therapies in a timely manner in post-ischemic brain. Natural antioxidants [3] are able to activate vitagenes such as heat shock protein 70, heme oxygenase, thioredoxin reductase and sirtuins, which are part of the integrated cell stress tolerance system [181]. Thus, activation of the vitagene system, with elevation of thioredoxin reductase, heme oxygenase 1, glutathione and sirtuins, results in a reduction of pro-oxidative conditions [181]. Dietary antioxidants have been shown to have neuroprotective effects by activating hormetic pathways, including vitagene. Therefore, regulation of endogenous cellular defense processes through the heat shock system, natural and pharmacological antioxidant substances may be an interesting therapeutic approach in ischemic neurodegeneration of the brain. Thus, by sustaining or restoring vitagenes activity, it is possible to delay the development of post-ischemic injury and thereby increase life expectancy [180]. Therefore, preventing oxidative damage to DNA in brain cells and restoring gray and white matter and normal blood-brain barrier activity is critical to the successful treatment of post-ischemic brain.

## Figures and Tables

**Figure 1 antioxidants-11-00146-f001:**
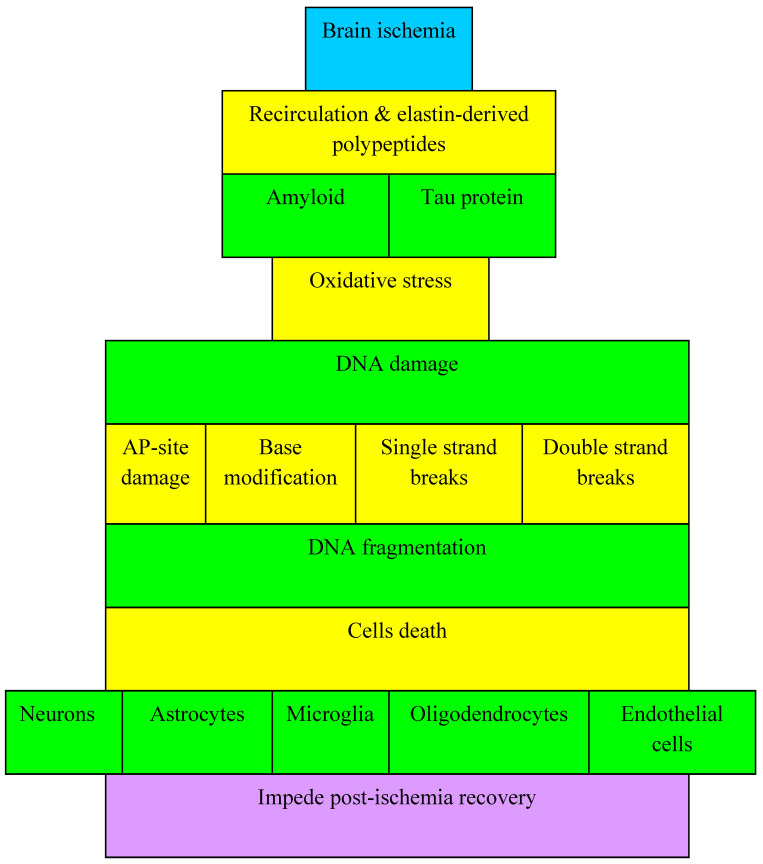
Cross talks between amyloid, tau protein and oxidative stress during DNA injury and fragmentation and brain cells death in post-ischemic brain.

**Figure 2 antioxidants-11-00146-f002:**
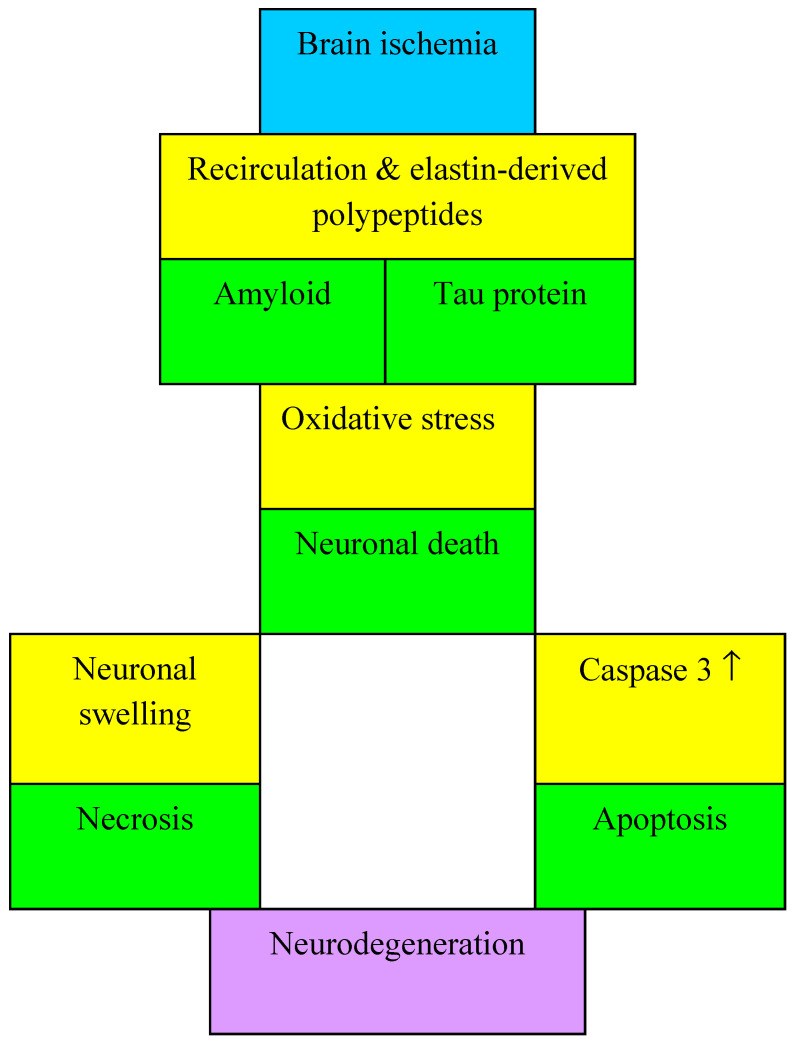
The interplay among oxidative stress, amyloid and tau protein in development necrosis, apoptosis and neurodegeneration. ↑—increase.

**Figure 3 antioxidants-11-00146-f003:**
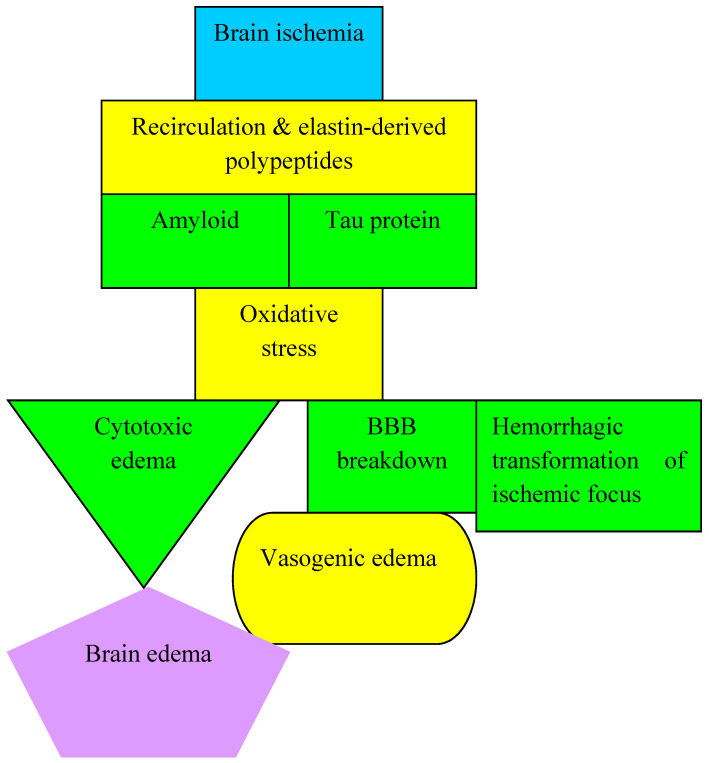
Complications in brain after post-ischemic recirculation. BBB-blood-brain barrier.

**Figure 4 antioxidants-11-00146-f004:**
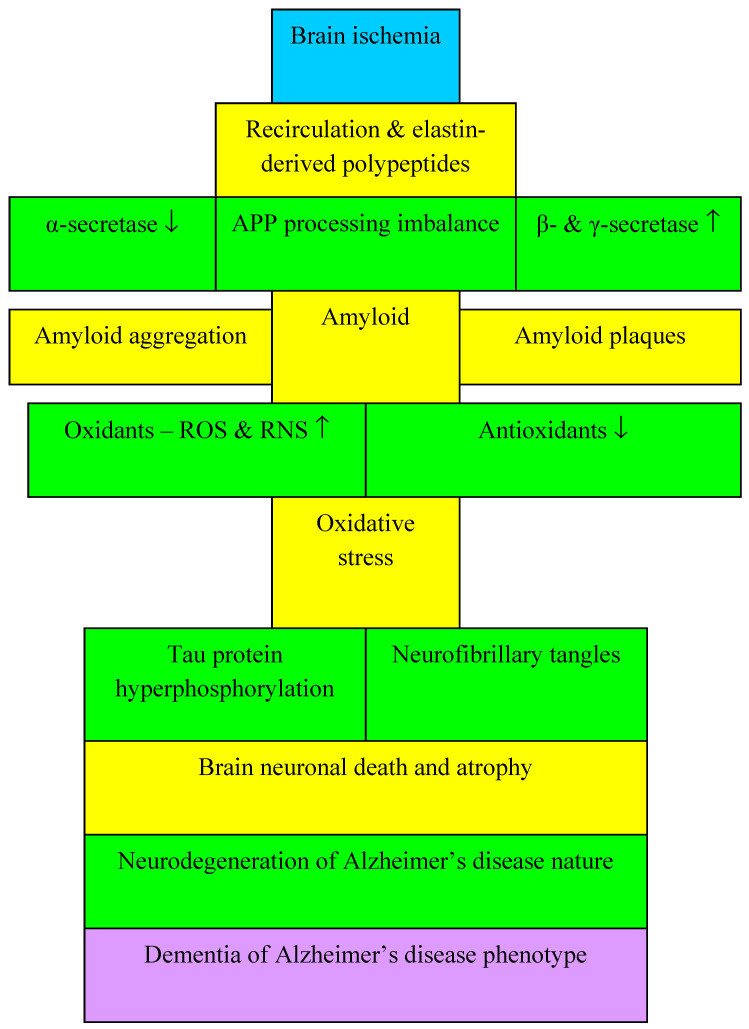
The interplay among amyloid, tau protein and oxidative stress during post-ischemic brain neurodegeneration of Alzheimer’s disease proteinopathy. ↓—decrease, ↑—increase, APP-amyloid protein precursor, ROS-reactive oxygen species, RNS-reactive nitrogen species.

**Table 1 antioxidants-11-00146-t001:** Post-ischemic expression of genes associated with Alzheimer’s disease in the CA1 and CA3 regions of the hippocampus.

	Genes	*APP*	*ADAM10*	*BACE1*	*PSEN1*	*PSEN2*	*MAPT*
Survival							
CA1 area							
2 days		↓	n.a.	↑↑	↑	↑↑	↑↑
7 days		↑	n.a.	↑	↑	↑	↑
30 days		↑	n.a.	↓	↓	↓	↓
CA3 area							
2 days		↔	↓	↓	↑	↔	↔
7 days		↑	↓	↓	↑	↓	↑
30 days		↔	↓	↑	↔	↑	↑

Expression: ↑ increase; ↑↑ increase; ↓ decrease, ↔ oscillation around control values; n.a. not available. Genes: *APP*-*amyloid protein precursor*, *BACE1*-*β-secretase*, *PSEN1*-*presnilin 1*, *PSEN2*-*presenilin 2*, *MAPT-tau protein*.

**Table 2 antioxidants-11-00146-t002:** Post-ischemic expression of genes associated with Alzheimer’s disease in the medial temporal cortex.

	Genes	*APP*	*BACE1*	*PSEN1*	*PSEN2*
Survival					
2 days		↓	↑↑	↔	↑↑
7 days		↑	↔	↔	↔
30 days		↑	↔	↔	↔

Expression: ↑ increase; ↑↑ increase; ↓ decrease, ↔ oscillation around control values. Genes: *APP*-*amyloid protein precursor*, *BACE1*-*β-secretase*, *PSEN1*-*presnilin 1*, *PSEN2*-*presenilin 2*.

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
