# Peer review of "Cross-Talk between Amyloid, Tau Protein and Free Radicals in Post-Ischemic Brain Neurodegeneration in the Form of Alzheimer’s Disease Proteinopathy"

_antioxidants, 2022, doi:10.3390/antiox11010146_

Round 1
Reviewer 1 Report
Review of the article entitled: Cross-talk between amyloid, tau protein and free radicals in postischemic brain neurodegeneration in the form of Alzheimer's disease proteinopathy.
In general, the manuscript is well prepared but has serious shortcomings. The authors ignore the cause of amyloid formation - why? Currently, the cause of β-amyloid formation is known, but the Authors completely ignored it, which makes the work obsolete and incomplete. Two independent research teams from China and Poland have clearly shown, that the cause of AD disease is the natural degradation of elastin. I found articles that clearly indicate that, elastin derived peptides increase in ROS level in brain (bot in in vivo and in vitro models) and causes AD development with β-amyloid formation.
Chapter 3 - Amyloid accumulation in post-ischemic brain. It is now known that after-ischemia, elastin derived peptides (EDPs) in brain are formed which induce formation of β-amyloid. Authors ignored research from the last 3 years why? The purpose of a review article is to summarize the current state of knowledge.
Chapter 2 - EDPs cause an increase in the amount of ROS in the brain in vivo and in vitro (in neural stem cells and astrocytes). Then EDPs cause production of β-amyloid in vitro and in vivo. The Authors may be surprised to find that the amount of EDPs in the brain increases with age and fully correlates with the occurrence of AD.
The authors still have a lot to learn, especially in the LATEST RESEARCH. On all diagrams after ischemia (before amyloid) should be added elastin derived peptides.
It has now been shown that EDPs are the cause of AD and they induce the formation of amyloid in in vivo models (rats, mice) and in vitro models (SH-SY5Y cell line, astrocytes and stem cells). The aforementioned data are crucial and must be taken into account because this is the current state of knowledge.
Author Response
Reviewer 1. All changes are in red. Review of the article entitled: Cross-talk between amyloid, tau protein and free radicals in postischemic brain neurodegeneration in the form of Alzheimer's disease proteinopathy. In general, the manuscript is well prepared but has serious shortcomings. The authors ignore the cause of amyloid formation - why? Currently, the cause of β-amyloid formation is known, but the Authors completely ignored it, which makes the work obsolete and incomplete. Two independent research teams from China and Poland have clearly shown, that the cause of AD disease is the natural degradation of elastin. I found articles that clearly indicate that, elastin derived peptides increase in ROS level in brain (bot in in vivo and in vitro models) and causes AD development with β-amyloid formation. --We have added 4 articles focusing on the problem presented by the reviewer (19-22). Thank you for your attention. Chapter 3 - Amyloid accumulation in post-ischemic brain. It is now known that after-ischemia, elastin derived peptides (EDPs) in brain are formed which induce formation of β-amyloid. Authors ignored research from the last 3 years why? The purpose of a review article is to summarize the current state of knowledge. --Done. Chapter 2 - EDPs cause an increase in the amount of ROS in the brain in vivo and in vitro (in neural stem cells and astrocytes). Then EDPs cause production of β-amyloid in vitro and in vivo. The Authors may be surprised to find that the amount of EDPs in the brain increases with age and fully correlates with the occurrence of AD. --Done. The authors still have a lot to learn, especially in the LATEST RESEARCH. On all diagrams after ischemia (before amyloid) should be added elastin derived peptides. --Done. It has now been shown that EDPs are the cause of AD and they induce the formation of amyloid in in vivo models (rats, mice) and in vitro models (SH-SY5Y cell line, astrocytes and stem cells). The aforementioned data are crucial and must be taken into account because this is the current state of knowledge. --Done.
Reviewer 2 Report
The paper entitled “Cross-talk between amyloid, tau protein and free radicals in post-ischemic brain neurodegeneration in the form of Alzheimer's disease proteinopathy “by Ryszard Pluta et al. is a review paper within the scope in Antioxidants. This review will present one of the newly characterized mechanisms that emerged with genomic and proteomic development that led to brain ischemia to a new level of post-ischemic neuropathological mechanisms, such as the presence of various amyloid plaques and the development of neurofibrillary tangles, which further exacerbate oxidative stress. I read the manuscript and felt it needs major modifications before considering it in this journal.
Please see my comments in the attached file.

Author Response
Reviewer 2. All changes are in red The paper entitled “Cross-talk between amyloid, tau protein and free radicals in post-ischemic brain neurodegeneration in the form of Alzheimer's disease proteinopathy “by Ryszard Pluta et al. is a review paper within the scope in Antioxidants. This review will present one of the newly characterized mechanisms that emerged with genomic and proteomic development that led to brain ischemia to a new level of post-ischemic neuropathological mechanisms, such as the presence of various amyloid plaques and the development of neurofibrillary tangles, which further exacerbate oxidative stress. I read the manuscript and felt it needs major modifications before consider in this journal. --Modifications have been made. Comments and suggestions Abstract: Abstracts should be self-contained and concise, explaining your work as briefly and clearly as possible. --Done. Introduction 1. With reference 1 add following citation. https://doi.org/10.3389/fphar.2020.619024 3.Rahman MH, Akter R, Bhattacharya T, Abdel-Daim MM, Alkahtani S, Arafah MW, Al-Johani NS, Alhoshani NM, Alkeraishan N, Alhenaky A, Abd-Elkader OH, El-Seedi HR, Kaushik D, Mittal V. Resveratrol and neuroprotection: Impact and its therapeutic potential in Alzheimer's disease. Front Pharmacol. 2020 Dec 30;11:619024. -- We added this work to the article (3). 2. Line127-130, the results show a progressive large redox imbalance due to the overproduction and accumulation of oxidative factors in the cells and brain tissue, which exceeds the clearance capacity by the antioxidant system. It’s a review article, Please check this sentence has written correctly. --Sentence is correct. 3. With reference 25, 26 add following citation. https://doi.org/10.3390/ijms222413313. --Jeong ES, Bajgai J, You IS, Rahman MH, Fadriquela A, Sharma S, Kwon HU, Lee SY, Kim CS, Lee KJ. Therapeutic effects of hydrogen gas inhalation on trimethyltin-induced neurotoxicity and cognitive impairment in the C57BL/6 mice model. Int J Mol Sci. 2021 Dec 10;22(24):13313. --Sorry. This article is not about post-ischemic brain changes. 4. Line 153-157, According to previous studies, overproduction of reactive oxygen species involves four pathways: NADPH oxidase, mitochondrial respiratory chain dysfunction, xanthine and hypoxanthine reaction mediated by xanthine oxidase and arachidonic acid catalyzed by cyclooxygenase 2.Please check these sentence has written correctly. --Checked and change. 5. These products are 4-hydroxynonenal, malondialdehyde and acrolein [2]. Typos error check and confirm. --It is OK! 6. Line 193-197, Protein 53 is a key moleculein the process of generating reactive oxygen species during the death of neurons in the brain following ischemia [2, 39,40]. Reactive oxygen species can cause a protein 53 reaction with cyclophilin D, which opens the mitochondrial transition pores, causing mitochondrial swelling and cells necrosis (Figure 2) [2,41]. What do u mean protein 53, please check and confirm this sentence has written correctly. --Sentence is OK! 7. Line 197-201, Reactive oxygen species increase the permeability of the mitochondrial membrane and affect the released cytochrome c, which forms an inhibitory complex with proteins of the Bcl-2 family, such as Bid and Bax, and proteins 53. Cytochrome c can generate active caspases by forming a complex with pro-caspase-9, an apoptotic protease activating factor-1 and ATP, resulting in apoptosis of Brain ischemia Recirculation Amyloid Tau protein Oxidative stress Neuronal death ï‚Caspase 3 Necrosis Apoptosis Neurodegeneration Neuronal swelling Antioxidants 2021, 10, x FOR PEER REVIEW 7 of 23 remodel neurons in the CA1 region of the hippocampus following reversible global cerebral ischemia in rats (Figure 2). Please check this sentence and write it correctly. --All is correct. 8. Line 221-223; also, reactive oxygen species can activate inflammatory cells. Reactive oxygen species activate neutrophils, microglia and macrophages by the nuclear factor kappa B pathway. Please check this sentence is repetition with introduction. --I checked and it is not repetition. 9. Section 2.4 with references add following references; https://doi.org/10.3390/biom11030350. --Arya A, Chahal R, Rao R, Rahman MH, Kaushik D, Akhtar MF, Saleem A, Khalifa SMA, El-Seedi HR, Kamel M, Albadrani GM, Abdel-Daim MM, Mittal V. Acetylcholinesterase Inhibitory Potential of Various Sesquiterpene Analogues for Alzheimer's Disease Therapy. Biomolecules. 2021 Feb 25;11(3):350 -- Sorry. This article does not deal with cerebral ischemia and recirculation. 10. Line 263-265, In addition, the oxidant peroxynitrite reacts with DNA elements, such as the sugar phosphate backbone and guanine nucleotides, causing DNA damage and ultimately activating PARP processes [2]. If first time, abbreviate it. --Done. 11. Line 281-284, The expression of the amyloid protein precursor (APP) gene in the CA1 area of the hippocampus decreased within 2 days, in contrast, it increased 7-30 days after ischemia (Table 1) [63]. Expression of the β-secretase gene (BACE1) increased during 2-7 days and decreased 30 days following ischemia (Table 1) [63]. Why it’s italic? --Genes are spelled italic. 12. Line 316-319, These data indicate that 316 after experimental ischemia, amyloid production is responsible for additional neurodegenerative processes that may worsen post-ischemic neurological outcomes due to the 318 constant wasting of neurons [6, 770, 71, 80, 81, 94, 95]. It has written incorrectly. --Done. 13. Line 352-354, After generating the sulfuranyl free radical, it can detach allylic H atoms from the acyl chains of unsaturated fatty acids in the lipid bilayer to initiate the lipid peroxidation process and, consequently, affect the lipid bilayer [101]. Please check this sentence has written correctly. --It is OK. 14. Line 364-366, Recent studies have documented an association between the expression of the tau protein (MAPT) gene in the CA1 and CA3 regions of the hippocampus after ischemia with 366 a 30-day survival (Table 1) [64]. Why Italic? --Genes are spelled italic. 15. Line 379-380, An increase in blood tau protein level was observed after global cerebral ischemia in 379 humans with a maximum increase on days 2 and 4 of survival [119]. Please check this sentence has written correctly. --It is OK. 16. Section 6, with reference add following references; https://doi.org/10.1016/j.exger.2021.111352. --Bhattacharya T, Dey PS, Akter R, Kabir MT, Rahman MH, Rauf A. Effect of natural leaf extracts as phytomedicine in curing geriatrics. Exp Gerontol. 2021 Jul 15;150:111352. Sorry. This article paper is not connected with tau protein versus oxidative stress in recirculation after brain ischemia. Conclusion: Your conclusion is so repetitive. --Changed Summarize the main ideas. --Done. Include a closing sentence. --Done. General comments: Lot of typos and grammatical errors throughout of the manuscript that should be revised with help of native speaker or professional scientific writer. This is few examples of your manuscript. You should revise your manuscript by the help of profession scientific writer (Native speaker) with English editing certificate. --Done.
Reviewer 3 Report
Interplay and coordination of redox interactions within endogenous antioxidant defence systems is an emerging area of reserach interest in anticancer and antidegenerative therapeutics. Mitochondria are key organelles involved in cellular survival, differentiation, and death induction. In this regard, mitochondrial morphology and/or function alterations are involved in stress-induced adaptive pathways, priming mitochondria for NO-driven neurotoxicity, mitophagy or apoptosis induction. As Nrf2 is a master regulator of the oxidative stress response inducing antioxidant-encoding gene expression, it is conceivable to hypothesize that antioxidants can increase the cellular resistance to oxidative stress by Nrf2-dependent mechanisms, in a process mediated by mitochondrial ROS (mtROS). Moreover, particular attention has been given to providing an assessment of the quantitative features of the dose-response relationships and underlying mechanisms that could account for the biphasic nature of the hormetic response after exposure to redox active agents, such as free radical oxygen species and their impact in inflammatory/antinflammatory pathways. The hormetic dose response should be seen as a reliable feature of the dose response for oxygen free radicals and their redox regulated transcriptional factors as well as antioxidant compounds and appears to have an important impact on brain pathophysiology and stress resistance mechanisms to oxidative and inflammatory insult and neurodegenerative damage. This is an interesting paper. The study is well-conceived and well-executed. This reviewer is satisfied with the significance of this study, the care in which the study was performed, and the implications of the results for human health. However, although the results presented are convincing, the work raises some concerns which will need to be addressed. The questions posed are of extremely high interest, but the paper does not give adequate definitive information, therefore pending addressing some major question is possible to accept for publication.
Major points
- Given the relationship between redox status and its possible biological relevance in neuroprotection, Authors while interpetrating results should discuss appropriately this aspect and make proper connection with emerging principles of hormesis. Indeed, preconditioning signal leading to cellular protection through Hormesis is an important redox dependent aging-associated to free radicals species accumulation, inflammatory responses involved in neurodegenerative/ neuroprotective mechanisms.
- Authors while interpetrating results should make proper connection with emerging principles of cellular stress response and vitagene network in coferring resilience phenotype. Indeed, preconditioning signal leading to cellular protection occurs via Vitagene brain axis activation and is an important redox dependent regulator of free radiacl species effects from physiological signaling to physiopathological consequences.
Author Response
Reviewer 3. All changes are in red. Interplay and coordination of redox interactions within endogenous antioxidant defence systems is an emerging area of reserach interest in anticancer and antidegenerative therapeutics. Mitochondria are key organelles involved in cellular survival, differentiation, and death induction. In this regard, mitochondrial morphology and/or function alterations are involved in stress-induced adaptive pathways, priming mitochondria for NO-driven neurotoxicity, mitophagy or apoptosis induction. As Nrf2 is a master regulator of the oxidative stress response inducing antioxidant-encoding gene expression, it is conceivable to hypothesize that antioxidants can increase the cellular resistance to oxidative stress by Nrf2-dependent mechanisms, in a process mediated by mitochondrial ROS (mtROS). Moreover, particular attention has been given to providing an assessment of the quantitative features of the dose-response relationships and underlying mechanisms that could account for the biphasic nature of the hormetic response after exposure to redox active agents, such as free radical oxygen species and their impact in inflammatory/antinflammatory pathways. The hormetic dose response should be seen as a reliable feature of the dose response for oxygen free radicals and their redox regulated transcriptional factors as well as antioxidant compounds and appears to have an important impact on brain pathophysiology and stress resistance mechanisms to oxidative and inflammatory insult and neurodegenerative damage. This is an interesting paper. The study is well-conceived and well-executed. This reviewer is satisfied with the significance of this study, the care in which the study was performed, and the implications of the results for human health. However, although the results presented are convincing, the work raises some concerns which will need to be addressed. The questions posed are of extremely high interest, but the paper does not give adequate definitive information, therefore pending addressing some major question is possible to accept for publication. --Thank you for drawing your attention to an important point, we now present a discussion on the suggested topic at the end of the conclusions. Major points 1. Given the relationship between redox status and its possible biological relevance in neuroprotection, Authors while interpetrating results should discuss appropriately this aspect and make proper connection with emerging principles of hormesis. Indeed, preconditioning signal leading to cellular protection through Hormesis is an important redox dependent aging-associated to free radicals species accumulation, inflammatory responses involved in neurodegenerative/neuroprotective mechanisms. 2. Authors while interpetrating results should make proper connection with emerging principles of cellular stress response and vitagene network in coferring resilience phenotype. Indeed, preconditioning signal leading to cellular protection occurs via Vitagene brain axis activation and is an important redox dependent regulator of free radiacl species effects from physiological signaling to physiopathological consequences. Ad. 1 and 2. We addressed the issues raised in the conclusions. Moreover, we have added 3 publications on the above subject (180-182).
Round 2
Reviewer 1 Report
The authors took into account a satisfactory number of my suggestions.
Reviewer 2 Report
The authors addressed all my comments. Now, I think it can be accepted.
Reviewer 3 Report
Accept